# Assessment of Risk Factors Associated with Severe Endometriosis and Establishment of Preoperative Prediction Model

**DOI:** 10.3390/diagnostics12102348

**Published:** 2022-09-28

**Authors:** Yanhua Yang, Jing Li, Hui Chen, Weiwei Feng

**Affiliations:** Department of Obstetrics and Gynecology, Ruijin Hospital, Shanghai Jiaotong University School of Medicine, 197 Ruijin Er Road, Huangpu District, Shanghai 200025, China

**Keywords:** endometriosis, ASRM staging, prediction model

## Abstract

Approximately 10% (176 million) of women of reproductive age worldwide suffer from endometriosis, which has a high rate of postoperative recurrence. The objective of this study was to investigate the risk factors of severe endometriosis and establish a preoperative prediction model. A retrospective analysis of a database established between January 2020 and March 2022 including 491 women with a pathology-based endometriosis diagnosis was conducted. Subjects were divided into two groups: the non-severe group (ASRM ≤ 40) and the severe group (ASRM > 40). Age ≥ 40 years, bilateral lesions, pelvic nodules, adenomyosis, APTT, CA125 ≥ 34.5 U/mL, D-dimer ≥ 0.34 mg/L, and maximum cyst diameter ≥ 58 mm were independent correlation factors for severe endometriosis. The logistic regression equation for these factors showed good diagnostic efficiency (AUC = 0.846), which was similar to the model with intraoperative indicators (AUC = 0.865). Patients with severe endometriosis also had a shorter APTT and higher D-dimer and PLT, indicating hypercoagulability. In conclusion, we constructed a simple and feasible formula involving parameters that are preoperatively accessible to predict the severity of endometriosis. This study is of reference value for determining the timing of and alternatives to surgery. At the same time, attention should be paid to the primary prevention of venous thrombosis and cardiovascular metabolic diseases in patients with severe endometriosis.

## 1. Introduction

Endometriosis is characterized by the presence, growth, and infiltration of endometrial tissue, which is typically considered to be formed of glands and stromata. These tissues occur outside the endometrium and myometrium and lead to pain, infertility, and nodules or masses [1]. It is estimated that 176 million women globally, accounting for 10% of women of reproductive age, suffer from endometriosis. The likelihood of continued or recurrent endometriosis is around 20% for patients within 2 years after surgery, soaring to 40–50% within 5 years after surgery [2]. Additionally, infertility occurs in up to 50% of women with endometriosis [3]. It is a chronic and progressive disease that can significantly affect a woman’s personal and professional life. Furthermore, we should highlight the need for customized and integrated treatment for these patients, considering both the physical and psychological aspects of their symptoms [4,5]. Although the social expense of endometriosis is as high as USD 80 billion per year globally, the awareness of endometriosis is quite low compared with other diseases that present a similar prevalence and social challenge [6]. Endometriosis is related to the backflow of menstrual blood, sex hormones, immunity, inflammation, genetics, and surgery (including cesarean deliveries or abdominal surgeries), etc., but its pathogenesis is still unclear [7].

Transvaginal sonography (TVS) is the most widely used imaging tool to diagnose endometriosis (93% sensitivity and 95% specificity) [3]. Laparoscopy is the method used to confirm endometriosis. It allows the exploration of the location and scope of the lesion and the acquisition of a biopsy for histopathological diagnosis [7]. At present, the American Society for Reproductive Medicine (ASRM) system is the global standard for endometriosis staging [8]. Briefly, the scoring criteria include the size and depth of pathological changes in the peritoneum and ovary, the extent and degree of adhesions on the ovary and fallopian tube, and the extent of adhesions on the uterine rectal cul de sac. Women with a more severe disease as scored by ASRM (III and IV) are at risk of all kinds of adverse obstetric-related outcomes [3,9]. A high ASRM score is also associated with a high recurrence rate and a low fertility index [10]. It has been shown that laparoscopy can increase pregnancy rates by removing endometriosis patches, but this is less likely to work in moderate to severe endometriosis [7].

It is clear that the prognosis of mild and severe endometriosis differ considerably. Mathew Leonardi et al. used preoperative ultrasound reports and surgical operative notes to retrospectively assign an ASRM score and stage. They concluded that ultrasound had high accuracy for predicting the mild, moderate, and severe ASRM stages of endometriosis [11]. However, the diagnostic performance of ultrasound depended on the expertise of the operator, as superficial endometriosis and fallopian tube adhesions are hardly visible via TVS. Both of these issues call into question the generalizability of their findings.

In this study, we intended to predict the severity of endometriosis according to common parameters of preoperative evaluation and to tailor interventions by accurately selecting an appropriate operation time, method, and scope. Our aim was to help patients prevent the recurrence of endometriosis and improve their fertility index and quality of life.

## 2. Materials and Methods

### 2.1. Subjects

This study was based on a retrospective analysis of a database prospectively established from January 2020 to March 2022 in Ruijin Hospital, Shanghai, China. All patients underwent laparoscopic surgery and were confirmed as having an endometrial cyst of the ovary according to pathology. All participants met the following inclusion criteria: non-pregnant; ≥18 years old; normal hepatic and renal function; a surgical indication for endometriosis or other benign gynecologic diseases; complete clinical, pathological, and sonographic imaging data of high quality. Exclusion criteria: pregnant or postmenopausal women; history of adnexectomy (ovary or fallopian tube), hysterectomy, or rectal resection; surgical history of severe pelvic infectious diseases such as pelvic tuberculosis, gangrene, or perforated appendicitis; women with coagulation disorders, autoimmune diseases, acute-phase infectious diseases, a diagnosis of uterine or ovarian malignancy, or the concomitant use of antiplatelet or anticoagulant therapy at the time of surgery; the administration of GnRH agonist ≥ 3 months before surgery; or ovulation induction one month before surgery.

### 2.2. Sonography

All patients underwent preoperative ultrasonography using Voluson E10 (GE Healthcare, Chicago, IL, USA), EIPQ5, and iU22 (Philips Healthcare, Manchester, UK) ultrasound machines with 5.0–9.0 MHz and 4.0–8.0 MHz transvaginal probes and 1.0–5.0 MHz transabdominal probes. Ultrasound parameters were adjusted to optimize the image quality for each patient. Traditionally, a routine pelvic ultrasound evaluates the uterus and ovaries, which leads to the detection of primary lesions [12]. Otherwise, as suggested by the International Deep Endometriosis Analysis (IDEA) consensus [13], endometriosis nodules in the anterior and posterior pelvic compartments are assessed. Multidimensional and multiangle real-time scans were performed. The location, size, shape, echogenic characteristics, and relationship with surrounding organs of the tumor were recorded. Color Doppler flow imaging within each tumor was also recorded and was graded according to the standard established by D. Timmerman et al. [14], i.e., no color flow signal, only minimal color signals, moderate color signals, or abundant color signals detected. When multiple adnexal masses were detected, we analyzed the mass with the most complex ultrasonographic morphology, and when masses had similar morphological characteristics, we chose the largest mass [15] (Figure 1).

### 2.3. Laboratory Tests

CA125 levels were measured using a chemoluminescence technique and an automatic analyzer (BECKMAN DXI800). Reference range: 0~23 U/mL.

A COULTER STKS fully automatic haemacytometer analyzer was used for blood platelet (PLT) and hemoglobin (Hb) quantification. Reference range: 100~300 × 10^9^/L and 120~160 g/L, respectively.

Coagulation parameters were detected by a Sysmex CS5100 automatic blood coagulation analyzer. Reference range: prothrombin time (PT), 10.0~16.0 s; activated partial thromboplastin time (APTT), 22.3~38.7 s; thrombin time (TT), 14.00~21.00 s; fibrinogen (Fg), 1.8~3.5 g/L; fibrin/fibrinogen degradation products (FDP), 0~5.0 mg/L; D-dimer, 0~0.55 mg/L.

### 2.4. Stages of Laparoscopic Surgery for Endometriosis

The revised scoring system of the American Society for Reproductive Medicine (ASRM) was applied to determine the stage of endometriosis on the basis of the size and depth of peritoneal and ovarian lesions, the scope and extent of ovarian and fallopian tube adhesions, and the degree of posterior cul-de-sac obliteration [8]. The scores were divided into four stages as follows: stage I (minimal), score 1–5; stage II (mild), score 6–15; stage III (moderate), score 16–40; stage IV (severe), score > 40.

In this study, stage I–III (score ≤ 40) was regarded as the non-severe group (group A), and stage IV (score > 40) was regarded as the severe group (group B). The definitive method for diagnosing peritoneal and deep endometriosis (DE) was visualization at surgery (typically involving laparoscopy).

### 2.5. Pathology

Pathology was the reference standard to determine the presence of endometrioma cysts and adenomyosis. Tissue specimens obtained during surgery were analyzed by a team of pathologists who specialized in gynecological pathology and were unaware of the ultrasound findings.

### 2.6. Statistical Analyses

Statistical analysis was performed using IBM SPSS Statistics, Version Chicago 23.0. Data normality was verified with the K–S test. Continuous variables were presented as mean ± SD for normal distribution or quartiles for non-normal distribution and were compared using the Student’s t test or the Mann–Whitney U test as appropriate. Categorical variables were presented as proportions and frequencies and were analyzed by the Chi-square test. ROC analysis was performed to determine the efficacy of different variables for distinguishing severe endometriosis, and the AUC value was calculated. Data yielding the maximal sum of sensitivity and specificity were set as the optimal critical values from ROC analysis. The Kappa consistency test was used to evaluate the congruence between preoperative ultrasound and intraoperative/pathological diagnosis. A binary logistic regression was conducted in order to evaluate the independent factors of severe endometriosis (the screening method was forward LR). Adjusted odds ratios with 95% CI were evaluated when a statistically significant correlation was found, and a regression equation was established. *p*-values < 0.05 were considered statistically significant.

## 3. Results

### 3.1. Baseline Characteristics of Subjects

The ASMR scores ranged from 13 to 148 with a median of 56. A total of 491 subjects, *n* = 165 (33.6%), were stage I–III (group A), whereas the remaining *n* = 326 (66.4%) were stage IV (group B). The patients’ ages ranged from 18 to 53 years old with a median of 37 years old. Due to the exclusion of adolescent and postmenopausal women, the age distribution was non-normal. Women in group B were older than group A (37 years vs. 34 years, *p* < 0.001), and the proportion of women ≥ 40 years old in group B was higher than in group A (41.1% vs. 22.4%, *p* < 0.001).

The number of gravidities ranged from zero to seven, with a median of one; the number of parities ranged from zero to three, with a median of one. The number of gravidities or parities in group B was higher than in group A (*p* = 0.013, *p* = 0.009). Sixty-nine cases (14.1%) had a history of ovarian cystectomy, including fifty cases of endometrial cyst. Fifty-five cases in group B had a history of oophorocystectomy, which was higher than the fourteen cases in group A (16.9% vs. 8.5%, *p* = 0.012) (Table 1).

### 3.2. Intraoperative and Pathological Findings

The number of bilateral endometrial cysts confirmed by surgery and pathology was 187, while the remaining *n* = 304 were unilateral lesions, including 172 left-sided and 132 right-sided. There was no statistical difference between the number of left or right lateral lesions (χ^2^ = 0.618, *p* = 0.432). On hundred and fifteen patients (23.4%) with adenomyosis were pathologically diagnosed. There were more patients who had adenomyosis in group B than in group A (30.1% vs. 10.3%, *p* < 0.001). Three hundred and ten patients (63.1%) were intraoperatively diagnosed with peritoneal and deep endometriosis, the proportion of which was significantly higher in group B than in group A (71.8% vs. 46.1%, *p* < 0.001) (Table 2).

### 3.3. Preoperative Features of Severe Endometriosis

#### 3.3.1. Sonographic Parameters

According to the judging principle [15], 258 left-sided and 232 right-sided ovary cysts were analyzed. One case could not be categorized as left or right because the cyst was too large and both ovaries were invisible. This case was classified as a bilateral lesion. The number of bilateral endometrial cysts diagnosed by ultrasound preoperatively was 163, accounting for 33.2%, while the remaining *n* = 328 (66.8%) were unilateral lesions. There were more bilateral lesions in group B than in group A (45.7% vs. 8.5%, *p* < 0.001) according to the preoperative sonographic findings.

In terms of grey-scale features, the maximum cyst diameter was 15~194 mm, with a median of 55 mm. Compared with group A, cysts were larger in group B (57 mm vs. 52 mm, *p* = 0.013). The constituent ratio of cysts ≥4 cm was 76.4% in group A and 82.5% in group B, but no statistical difference was obtained (χ^2^ = 2.636, *p* = 0.104). A total of 265 unilocular cysts were found in 54.0% of all subjects, *n* = 103 (constituent ratio = 62.4%) in group A and *n* = 162 (constituent ratio = 49.7%) in group B. The difference exhibited statistical significance (*p* = 0.008). The number of cavities in each cyst ranged from 1 to 15, with a median of 1. Cavities in group B were more numerous than in group A (2 vs. 1, *p* = 0.007). Cavities in multilocular cysts between the two groups were further compared, and the results showed no statistical significance (Z= −0.619, *p* = 0.536).

Among the 301 cysts without color Doppler flow, accounting for 61.3% of the total subjects, there were 102 cases in group A (constituent ratio = 61.8%) and 199 cases in group B (constituent ratio = 61.0%), which showed no statistical significance (χ^2^ = 0.028, *p* = 0.868). There were three cases with abundant blood flow signals, accounting for 0.6%, including two cases in group A and one case in group B.

One hundred and sixty-eight cases of adenomyosis were diagnosed by ultrasound, accounting for 34.2%. The proportion of adenomyosis in group B was higher than that in group A (44.5% vs. 13.9%, *p* < 0.001). One hundred and ten cases of pelvic endometriosis nodules were diagnosed by preoperative ultrasound, accounting for 22.4%. The proportion in group B was significantly higher than that in group A (29.4% vs. 8.5%, *p* < 0.001).

#### 3.3.2. Laboratory Parameters

The coagulation parameters, PLT count, Hb level, and CA125 level are listed in Table 3. Compared with group A, group B exhibited higher CA125 levels (28.4 U/mL vs. 46.1 U/mL, *p* < 0.001), shorter APTTs (30.2 s vs. 29.7 s, *p* = 0.015), increased D-dimer and PLT levels (*p* < 0.001, *p* = 0.005), and decreased Hb levels (124 g/L vs. 121 g/L, *p* = 0.015) (Table 2).

### 3.4. Consistency between Preoperative Ultrasonic Findings and Surgical Pathological Diagnosis

In order to evaluate the capacity of ultrasound to predict severe endometriosis, we tested the consistency of the preoperative sonography indicators with intraoperative and pathological diagnosis. The results showed that the ultrasound results were in good agreement with intraoperative and pathological diagnosis for categorizing unilateral/bilateral lesions (Kappa = 0.805, *p* < 0.001). With regard to the diagnosis of adenomyosis, the consistency was general (Kappa = 0.623, *p* < 0.001). In terms of evaluating peritoneal and deep endometriosis, the consistency was poor (Kappa = 0.217, *p* < 0.001).

### 3.5. Receiver Operating Curve and Cutoff Value

Continuous variables with statistical differences between the two groups included age, the number of gravidities and parities, CA125, APTT, D-dimer, PLT count, Hb level, the maximum cyst diameter, and the number of cavities. ROC analysis was used to estimate the diagnostic efficiency of each index. The results showed that all the areas under curve (AUC) were less than 0.7 (Table 4), indicating poor diagnostic value. Next, we chose five indicators for cutoff value evaluation (three indicators with AUC> 0.6, i.e., age, CA125, D-dimer; and two sonographic indicators, i.e., the maximum cyst diameter and the number of cavities). Age ≥ 40 yrs, CA125 ≥ 34.5 U/mL, D-dimer ≥ 0.34 mg/L, maximum cyst diameter ≥ 58 mm, and multilocular cysts (≥2 cavities) were selected as the bases for the categorization. The chi-square test was carried out, and significant predictive values were retained: *p* < 0.001, *p* < 0.001, *p* < 0.001, *p* = 0.002, and *p* = 0.008, respectively (Table 1 and Table 2).

### 3.6. Independent Related Factors and Predicting Model of Severe Endometriosis

According to the above results, parameters with statistical differences between groups A and B were set as logistic regression covariates. Ten parameters including age ≥ 40 yrs, bilateral endometrial cysts diagnosed by ultrasound, adenomyosis diagnosed by ultrasound, pelvic endometriosis nodules diagnosed by ultrasound, bilateral lesions confirmed by surgery and pathology, the intraoperative finding of peritoneal and deep endometriosis, APTT, CA125 ≥ 34.5 U/mL, D-dimer ≥ 0.34 mg/L, and maximum cyst diameter ≥ 58 mm were independent related factors of severe endometriosis (Table 5). ROC analysis according to the predicted value of logistic regression showed that the AUC = 0.865 (95% CI: 0.834~0.897, *p* < 0.001).

Furthermore, we set preoperative parameters with statistical differences between groups A and B as logistic regression covariates, i.e., we removed three intraoperative or pathological parameters. The results demonstrated that eight indicators including age ≥ 40 yrs, bilateral endometrial cysts diagnosed by ultrasound, pelvic endometriosis nodules diagnosed by ultrasound, adenomyosis diagnosed by ultrasound, APTT, CA125 ≥ 34.5 U/mL, D-dimer ≥ 0.34 mg/L, and maximum cyst diameter ≥ 58 mm were independent related factors of severe endometriosis (Table 6). ROC analysis according to the predicted value of logistic regression showed that the AUC = 0.846 (95% CI: 0.811~0.880, *p* < 0.001) (Figure 2), the diagnostic efficiency of which was equivalent to the previous model including intraoperative parameters.

Therefore, the logistic regression equation was established as follows:Logit (P| outcome = severe endometriosis) = 0.718∙X1 + 2.264∙X2 + 1.251∙X3+ 1.257∙X4 − 0.127∙X5 + 0.797∙X6 + 0.793∙X7 + 0.571∙X8

X1 = age ≥ 40 yrs; X2 = bilateral endometrial cyst diagnosed by preoperative ultrasound; X3 = pelvic endometriosis nodules diagnosed by preoperative ultrasound; X4 = adenomyosis diagnosed by preoperative ultrasound; X5 = APTT (s); X6 = CA125 ≥ 34.5 U/mL; X7 = D-dimer ≥ 0.34 mg/L; and X8 = maximum cyst diameter ≥ 58 mm.

## 4. Discussion

There is often a long diagnostic delay (7~9 years) for endometriosis after the onset of symptoms [2]. The patient’s age reflects the course of disease, i.e., the older the patient is, the longer she may have suffered from endometriosis, regardless of whether she was diagnosed. It can be logically inferred that the longer the course of the disease, the more instances of ectopic endometrial bleeding with the onset of menstruation occur, leading to more extensive adhesions and higher ASRM scores. Accordingly, we found that patients in group B were older than those in group A. We also found that compared with group A, the proportion of subjects beyond 40 years old was higher in group B.Over 40 years of age is also one of the thrombosis risks for compound oral contraceptives (COCs) [16,17].

Greater parity has been reported to be associated with a lower risk of endometriosis [18]. In our study, the number of gravidities and parities was higher in the severe group. The different results may have been related to differences in the control group. Since the number of gravidities or parities is not an independent factor of severe endometriosis, the predictive value needs to be further discussed.

Although a definite diagnosis of endometriosis is based on histological criteria, the histological sampling depends on visual diagnosis. However, the visualization of some early lesions can be challenging due to the high heterogeneity of the locations and manifestations [3]. Therefore, it is necessary to focus on clinical diagnosis rather than surgical diagnosis alone for the early identification and treatment of endometriosis [19]. Endometriosis has a typical sonographic appearance, including homogenous low-level or ground glass internal echoes, though it has no specific symptoms and signs. Transvaginal sonography (TVS) is the preferred imaging tool. Although it is an effective method for diagnosing endometriomas (93% sensitivity and 95% specificity), TVS has a limited capability for diagnosing peritoneal lesions [20]. Some early peritoneal lesions only show follicles or pigmentation without nodular formation; therefore, they cannot be detected by ultrasound. This also explains the poor consistency between the preoperative ultrasound diagnosis and intraoperative diagnosis of peritoneal and deep endometriosis. However, the preoperative ultrasound diagnosis of endometriosis nodules was still an independent risk factor for the severe group (OR = 3.494). This was consistent with the finding that TVS improved the sensitivity and specificity of DE diagnosis [11,13,21,22].

In terms of the other ultrasound parameters, the maximum cyst diameter and presence of bilateral lesions were associated with severe endometriosis, and they are also scoring items in the ASRM system. We found no difference in the proportion of patients with endometrial cysts > 40 mm between the two groups. 40 mm is the cutoff value of surgical indication according to the guidelines [17]. In addition, a cutoff value of 58 mm (≈6 cm) was obtained by the ROC analysis. We found that the proportion of patients with cysts ≥ 58 mm in the severe group was significantly higher than that in the non-severe group. Although endometriomas are common in infertile women, whether they should be surgically removed before an in vitro fertilization (IVF) is still debated. Sufficient proof has not been presented to draw firm recommendations [23]. Evidence shows that simply removing unilateral endometrial cysts with a diameter of 6 cm does not significantly increase the spontaneous pregnancy rate. Additionally, there is still a lack of evidence on whether surgery can improve the pregnancy rate in patients with stage III–IV endometriosis [17]. All the available information suggests that the choice of therapy should be based on the comprehensive judgment of other indicators. The pros and cons of surgical risk, fertility improvement, and recurrence prevention should be weighed. A multidisciplinary endometriosis team is an important factor for achieving good surgical outcomes [24].

We also found that the proportion of multilocular cysts was higher in the severe endometriosis group compared with the non-severe group. This may have been related to the intracapsular hemorrhage of the endometrial cysts during the menstrual cycle, resulting in adhesions and fiber separation. However, multiple locules and the number of locules were not independent risk factors of severe endometriosis.

There has been much progress in developing various serum biomarkers as possible diagnostic tools. For example, the joint detection of multiple blood microRNAs became a research spot [25], although it was affected by poor repeatability [26]. Hormone receptor status has been used as a biomarker to predict therapeutic response. However, these biomarkers, whether used alone or in combination, have not achieved reliable diagnostic efficiency. CA125 is a surface antigen of coelothelium metaplastic tissue, mainly distributed on mesothelium cells and the gyneduct. We found that the quantitative level of CA125 in the severe endometriosis group was significantly higher than that in the non-severe group. This result was consistent with the results from several other studies [27,28,29]. We also found that CA125 ≥ 34.5 U/mL was an independent risk factor for a severe form of the disease, which could assist in judging the severity of endometriosis.

In terms of coagulation indicators, Li Ruobing et al. reported that the PT level in the stage III group was higher than in the stage IV group, while the FIB and PLT levels were lower than in the stage IV group [30]. Other studies also support the conclusion that women with endometriosis have hypercoagulability and fibrinolysis abnormalities [31,32,33]. Platelets, coagulation factors, and the fibrinolytic system are important components of the coagulation mechanism. We discovered that compared with the non-severe group, women with severe endometriosis had higher platelet counts, shorter APTTs, and higher levels of D-dimer. All the indices demonstrated a hypercoagulation status. The cause of this may have been the activation of the intrinsic coagulation system by the repeated bleeding of the ectopic endometrium during the menstrual cycle. Platelets participate in the process of coagulation through adhesion and aggregation mediated by membrane glycoproteins. It seems that the alterations to the coagulation parameters in patients with endometriosis are subtle and still within the normal range. However, Ding et al. found that the coagulation measurements were all significantly changed three months after the removal of endometriotic lesions, suggesting the possible role of active endometriosis in the disturbance of the local or systemic coagulation index [34]. Furthermore, it has been reported that patients with endometriosis have a high probability of systemic cardiovascular and metabolic diseases [3,35]. Therefore, we cannot rule out the possibility that these systemic perturbations may contribute to the pathogenetic process of the disease or to the increased cardiovascular and thrombotic morbidity observed in affected patients. The findings mentioned above remind us to pay attention to the coagulation state of endometriosis, especially in the severe stage. It is necessary to conduct a full evaluation before clinical intervention, such as with COCs or surgery, to guard against venous thrombosis and other adverse events. The primary prevention of cardiovascular and metabolic diseases should also be emphasized.

Platelets, fibrinogen, and D-dimer are involved in not only the coagulation system, but also systemic inflammation and the development of malignant diseases such as ovarian cancer. Rafał Watrowski et al. reported that PLT is a ubiquitously available parameter that could be used in the evaluation of pelvic masses. Thrombocytosis combined with CA125 improves the capacity for detecting adnexal malignancy [36]. Indeed, there is a clinical score that includes CA125, platelet count, and fibrinogen [37] to predict ovarian malignancy. The plasma D-dimer level in ovarian cancer patients has been proven to predict changes correlated with disease progression and VTE risk [38]. From the clinical point of view, disturbances in the coagulation parameters should be a cause for concern for every gynecologist as a possible indicator of both severe endometriosis and ovarian malignancy.

The decrease in Hb levels in patients with severe endometriosis may have been related to the higher prevalence of adenomyosis. Adenomyosis leads to increased menstrual flow and prolonged periods, which cause anemia. The incidence of adenomyosis in combination with endometriosis is 21.3~91.1% [17]. Nadine Di Donato et al. discovered that the prevalence of adenomyosis in patients with endometriosis was 21.8% [39], which was in accordance with our figure of 23.3%. In our study, the consistency between the preoperative ultrasound diagnosis of adenomyosis and surgical pathological results was not very satisfactory (Kappa = 0.624). We propose two possible reasons for this. First, the sonographic appearance of some adenomyomas is similar to that of a myoma, which leads to misdiagnosis. Second, some lesions are too small to be recognized by the naked eye, and thus pathological diagnosis cannot be achieved, resulting in bias.

The strength of this study was our development of a predictive model for severe endometriosis. The predictors are easy to obtain and provide additional insights into the pathogenesis of endometriosis. The limitation was that the study design was retrospective and did not adopt ENZIAN classification, which considers the involvement of retroperitoneal structures in DE. Compared with ENZIAN, ASRM staging demonstrates limited reproducibility in regard to the staging findings when the ovaries and posterior cul-de-sac are involved. However, the ASRM score is the most widely used staging system globally. Fortunately, the severity of endometriosis according to the ENZIAN classification and the ASRM score appears to correlate. Thus, the use of the ENZIAN classification could be recommended as a supplement to the ASRM score for the detailed description of endometriosis [40]. A further study including the ENZIAN classification is needed.

## 5. Conclusions

In conclusion, this study utilized routine clinical parameters to propose a formula for preoperatively predicting the severity of endometriosis. The system is simple and non-invasive and may be applied in clinical practice. However, considering the diagnostic bias and reverse causation in this retrospective study, prospective design and rigorous methods are needed to facilitate comparison and replication to advance our understanding of endometriosis. This study could also be a reference for tailoring interventions to improve the health of patients.

## Figures and Tables

**Figure 1 diagnostics-12-02348-f001:**
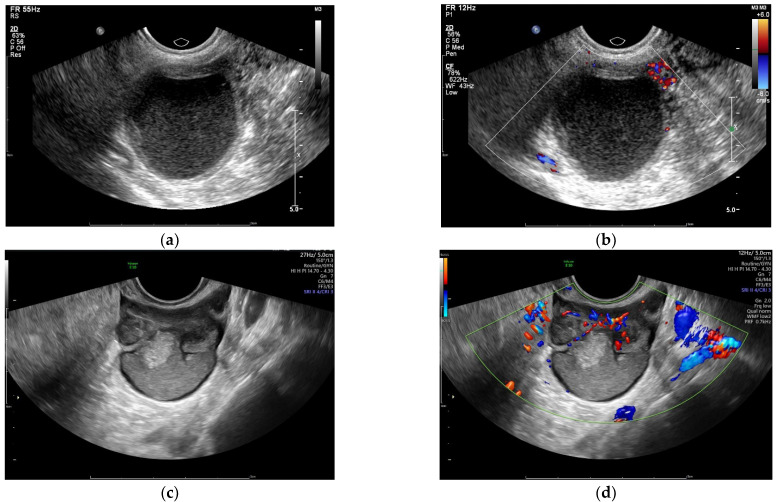
Sonographic images of endometriosis cysts confirmed by pathology: (**a**,**b**) unilocular cyst without color Doppler flow; (**c**,**d**) multilocular cyst/solid tumor with abundant color Doppler flow.

**Figure 2 diagnostics-12-02348-f002:**
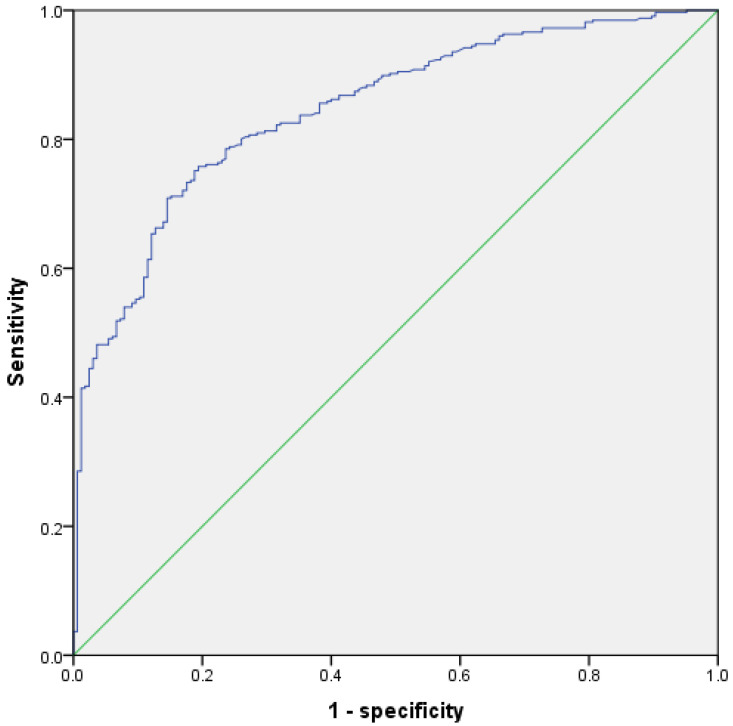
ROC of logistic regression model for predicting severe endometriosis with preoperative parameters.

**Table 1 diagnostics-12-02348-t001:** Baseline characteristics of 491 subjects.

Baseline Characteristics		Non-Severe Endometriosis (Group A, *n* = 165)	Severe Endometriosis (Group B, *n* = 326)	*p*-Value
Age * (years old)		34 (29~39)	37 (31~42)	<0.001
Age	≥40 years old	37 (22.4%)	134 (41.1%)	<0.001
	<40 years old	128 (77.6%)	192 (58.9%)	
Number of gravidities *		1 (0~1)	1 (0~2)	0.013
Number of parities *		0 (0~1)	1 (0~1)	0.009
History of ovarian cystectomy	Yes	14 (8.5%)	55 (16.9%)	0.012
	No	151 (91.5%)	271 (83.1%)	

* Quartile for non-normal distribution data.

**Table 2 diagnostics-12-02348-t002:** Preoperative and intraoperative/pathological features.

Features		Non-Severe Endometriosis (Group A, *n* = 165)	Severe Endometriosis (Group B, *n* = 326)	*p*-Value
Endometrial cysts **	Unilateral	151 (91.5%)	177 (54.3%)	<0.001
	Bilateral	14 (8.5%)	149 (45.7%)	
Adenomyosis **		23 (13.9%)	145 (44.5%)	<0.001
		142 (86.1%)	181 (55.5%)	
Pelvic endometriosis nodules **	Yes	14 (8.5%)	96 (29.4%)	<0.001
	No	151 (91.5%)	230 (70.6%)	
Endometrial cysts ***	Unilateral	146 (88.5%)	158 (48.5%)	<0.001
	Bilateral	19 (11.5%)	168 (51.5%)	
Adenomyosis ***	Yes	17 (10.3%)	98 (30.1%)	<0.001
	No	148 (89.7%)	228 (69.9%)	
Intraoperative finding of peritoneal and deep endometriosis	Yes	76 (46.1%)	234 (71.8%)	<0.001
	No	89 (53.9%)	92 (28.2%)	
Maximum cyst diameter	≥40 mm	126 (76.4%)	269 (82.5%)	0.104
	<40 mm	39 (23.6%)	57 (17.5%)	
Maximum cyst diameter	≥58 mm	57 (34.5%)	160 (49.1%)	0.002
	<58 mm	108 (65.5%)	166 (50.9%)	
Unilocular cyst	Yes	103 (62.4%)	162 (49.7%)	0.008
	No	62 (37.6%)	164 (50.3%)	
Color Doppler flow	Without	102 (61.8%)	199 (61.0%)	0.868
	With	63 (38.2%)	127 (39.0%)	
Number of cavities *		1 (1~2)	2 (1~3)	0.007
Maximum cyst diameter * (mm)		52 (40~67)	57 (43~72)	0.013
CA125 * (U/mL)		28.4 (17.8~53.3)	46.1 (25.4~84.6)	<0.001
CA125	≥34.5 U/mL	67 (40.6%)	208 (63.8%)	<0.001
	<34.5 U/mL	98 (59.4%)	118 (36.2%)	
APTT * (s)		30.2 (28.4~32.7)	29.7 (28.2~31.6)	0.015
PT * (s)		11.5 (11.0~12.1)	11.4 (11.0~12.0)	0.481
INR *		0.97 (0.93~1.03)	0.96 (0.93~1.02)	0.502
TT * (s)		18.5 (17.7~19.2)	18.5 (17.6~19.0)	0.607
Fg * (g/L)		2.5 (2.2~2.8)	2.5 (2.3~2.9)	0.295
FDP * (mg/L)		1.3 (1.2~1.7)	1.5 (1.2~2.1)	0.050
D-dimer * (mg/L)		0.28 (0.21~0.33)	0.31 (0.24~0.39)	<0.001
D-dimer	≥0.34 mg/L	38 (23.0%)	133 (40.8%)	<0.001
	<0.34 mg/L	127 (77.0%)	193 (59.2%)	
PLT * (×10^9^/L)		235 (194~277)	255 (216~293)	0.005
Hb * (g/L)		124 (116~131)	121 (112~128)	0.015

* Quartile for non-normal distribution data. ** Diagnosed by ultrasound. *** Diagnosed by surgery and pathology.

**Table 3 diagnostics-12-02348-t003:** Laboratory parameters of 491 subjects.

Parameter	Minimum	Maximum	P25	P50	P75
Ca125 (U/mL)	7.1	2657.2	22.0	38.8	72.2
APTT (s)	23.8	39.1	28.3	29.8	32.1
PT (s)	10.0	15.1	11.0	11.4	12.0
INR	0.84	1.29	0.93	0.96	1.02
TT (s)	15.00	21.10	17.60	18.50	19.20
Fg (g/L)	1.6	6.3	2.2	2.5	2.9
FDP (mg/L)	0.5	34.1	1.2	1.5	1.9
D-dimer (mg/L)	0.10	9.30	0.23	0.29	0.37
PLT (×10^9^/L)	89	564	206	247	287
Hb (g/L)	73	156	113	122	129

**Table 4 diagnostics-12-02348-t004:** ROC analysis of continuous variables with statistical differences between groups A and B.

Variables	AUC	*p*-Value	95% CI
Age	0.604	<0.001	0.552~0.657
Maximum cyst diameter	0.568	0.014	0.514~0.622
CA125	0.638	<0.001	0.586~0.690
Number of gravidities	0.565	0.019	0.512~0.618
Number of parities	0.565	0.014	0.512~0.617
Number of cavities	0.568	0.013	0.516~0.621
APTT	0.433	0.015	0.378~0.487
D-dimer	0.604	<0.001	0.552~0.656
PLT	0.577	0.005	0.523~0.632
Hb	0.433	0.015	0.380~0.486

**Table 5 diagnostics-12-02348-t005:** Independent related factors of severe endometriosis.

Variables	B	*p*-Value	OR	95% CI
Age ≥ 40 yrs	0.802	0.005	2.231	1.270~3.918
Bilateral endometrial cysts *	1.027	0.040	2.792	1.045~7.457
Pelvic endometriosis nodules *	1.131	0.002	3.097	1.505~6.375
Adenomyosis *	1.260	<0.001	3.525	1.942~6.399
Intraoperative finding of peritoneal and deep endometriosis	0.881	0.001	2.412	1.454~4.004
Bilateral endometrial cysts **	1.621	<0.001	5.059	2.072~12.354
APTT	−0.127	0.011	0.880	0.798~0.971
CA125 ≥ 34.5 U/mL	0.704	0.005	2.021	1.233~3.313
D-dimer ≥ 0.34 mg/L	0.840	0.002	2.317	1.371~3.918
Maximum cyst diameter ≥ 58 mm	0.634	0.012	1.885	1.153~3.083
Constant	−2.542	0.119	0.079	

* Diagnosed by ultrasound. ** Diagnosed by surgery and pathology.

**Table 6 diagnostics-12-02348-t006:** Preoperative independent related factors of severe endometriosis.

Variables	B	*p*-Value	OR	95% CI
Age ≥ 40 yrs	0.718	0.009	2.050	1.197~3.511
Bilateral endometrial cysts *	2.264	<0.001	9.624	5.053~18.331
Pelvic endometriosis nodules *	1.251	<0.001	3.494	1.768~6.906
Adenomyosis *	1.257	<0.001	3.517	1.978~6.252
APTT	−0.127	0.009	0.880	0.800~0.968
CA125 ≥ 34.5 U/mL	0.797	0.001	2.220	1.382~3.566
D-dimer ≥ 0.34 mg/L	0.793	0.002	2.210	1.334~3.661
Maximum cyst diameter ≥ 58 mm	0.571	0.018	1.770	1.104~2.839
Constant	−1.525	0.329	0.218	

* Diagnosed by ultrasound.

## Data Availability

No new data were created or analyzed in this study. Data sharing is not applicable to this article.

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
