# Peer review of "Assessment of Risk Factors Associated with Severe Endometriosis and Establishment of Preoperative Prediction Model"

_diagnostics, 2022, doi:10.3390/diagnostics12102348_

Round 1

Reviewer 1 Report

I read with great interest the manuscript, which falls within the aim of this Journal. In my honest opinion, the topic is interesting enough to attract the readers’ attention. Nevertheless, the authors should clarify some points and improve the discussion, as suggested below.

Authors should consider the following recommendations:

-       Manuscript should be further revised in order to correct some typos and improve style.

-        Authors should discuss how the main target of future algorithm should be the prediction of difficult surgery, especially in case of deep infiltrating endometriosis, as well as reproductive prognosis with or without surgery (authors may refer to: PMID: 34008386; PMID: 34585638).

Author Response

Thank you for your comments and suggestions. According to your advice, we revised our article as follows:

Point 1: Manuscript should be further revised in order to correct some typos and improve style.

Response 1: Manuscript was further revised and corrected some typos and improve style.

Point 2: Authors should discuss how the main target of future algorithm should be the prediction of difficult surgery, especially in case of deep infiltrating endometriosis, as well as reproductive prognosis with or without surgery (authors may refer to: PMID: 34008386; PMID: 34585638).

Response 2: According to the reference, we pointed out that the future target should be the prediction of difficult surgery, especially in case of deep infiltrating endometriosis, as well as reproductive prognosis in page 10 line 318~321 & line 327~328.

Reviewer 2 Report

I appreciate the authors' efforts to develop a predictive model for severe endometriosis. The laboratory predictors are easy to obtain and provide additional insight into the pathogenesis of endometriosis.  I see two limitations of the study that should be addressed in the discussion. 1. The authors use the American Society for Reproductive Medicine (ASRM) staging. However, this classification is of limited value in cases of deep/retroperitoneal endometriosis. The more appropriate classification system would probably be the ENZIAN classification, which includes involvement of retroperitoneal structures with deep infiltrating endometriosis. Fortunately, the severity of endometriosis according to the ENZIAN classification and the rASRM score appear to correlate (PMID: 23451860) so that a change in methodology is not required, but the objections should be addressed in the discussion. 2. The role of platelets and D-dimer is interpreted only as a disturbance within the coagulation system. In my opinion, this view is too one-dimensional. The observed changes (as important as they are used to build the model) may reflect the increased systemic inflammation, since platelets or fibrinogen are not also part of the coagulation system, but they are involved in the inflammation and the most important development of e.g. malignant diseases of the ovaries. There is even a clinical score including CA125, platelet count and fibrinogen (PMID: 26499778) to predict ovarian malignancy (the same parameters as those investigated in the present work).

3. From the clinical point of view, it shoud be addressed that elevated platelet count or disturbances in the coagulation parameters should be alarming for every gynecologists, not only as being possibly indicative for endometriosis, but also ovarian malignancy (PMID: 27207344, PMID: 26499778, PMID: 28640083).

Author Response

Thank you for your comments and suggestions. According to your advice, we revised our article as follows:

Point 1: The authors use the American Society for Reproductive Medicine (ASRM) staging. However, this classification is of limited value in cases of deep/retroperitoneal endometriosis. The more appropriate classification system would probably be the ENZIAN classification, which includes involvement of retroperitoneal structures with deep infiltrating endometriosis. Fortunately, the severity of endometriosis according to the ENZIAN classification and the rASRM score appear to correlate (PMID: 23451860) so that a change in methodology is not required, but the objections should be addressed in the discussion.

Response 1: ENZIAN classification includes involvement of retroperitoneal structures with deep infiltrating endometriosis and gives a relatively precise morphological description. However, in our retrospective research, gynecologists used ASRM staging according to the textbook or guideline. We have addressed the objections in the discussion and pointed out that the severity of endometriosis according to the ENZIAN classification and the rASRM score appears to correlate (PMID: 23451860) (page 11 line 397 ~ 400). In future study, we will take ENZIAN classification into account.

Point 2: The role of platelets and D-dimer is interpreted only as a disturbance within the coagulation system. In my opinion, this view is too one-dimensional. The observed changes (as important as they are used to build the model) may reflect the increased systemic inflammation, since platelets or fibrinogen are not also part of the coagulation system, but they are involved in the inflammation and the most important development of e.g. malignant diseases of the ovaries. There is even a clinical score including CA125, platelet count and fibrinogen (PMID: 26499778) to predict ovarian malignancy (the same parameters as those investigated in the present work).

Response 2: Thank you for recommending literature (PMID: 26499778) for reference, we cited it in our article (page 11 line 373 ~ 375 ) and discussed the involvement of coagulation parameters in inflammation and malignant diseases of the ovaries.

Point 3: From the clinical point of view, it shoud be addressed that elevated platelet count or disturbances in the coagulation parameters should be alarming for every gynecologists, not only as being possibly indicative for endometriosis, but also ovarian malignancy (PMID: 27207344, PMID: 26499778, PMID: 28640083).

Response 3: We addressed in discussion section that elevated platelet count or disturbances in the coagulation parameters should be alarming for every gynecologist, not only as being possibly indicative for endometriosis, but also ovarian malignancy (page 11 line 371 ~ 379 ). Recommended references were also cited.

Reviewer 3 Report

Dear Editor, 

thank you for the opportunity to review the manuscript entitled ''Assessment of risk factors associated with severe endometriosis and establishment of preoperative prediction model''. Although there are several facts that are original and may present a new direction in diagnostic workup of patients with endometriosis, I have some concerns to address. 

Introduction section has to be amended with few sentences about impaired quality of life of patients with endometriosis. Please cite two recent papers which studied this particular problem (PMID: 35819491 and PMID: 34718292). 

Methodology and Results are clearly presented although I believe that #ENZIAN classification will be of more interest due to its comprehensive nature. This is why I suggest that it is important to include this classification in Discussion section. Please compare its advantages and limitations over ASRM which is used in your study. 

Please clarify strengths and limitations of your study. 

Moderate English changes are required (grammar/spell check).  

Author Response

Thank you for your comments and suggestions. According to your advice, we revised our article as follows:

Point 1: Introduction section has to be amended with few sentences about impaired quality of life of patients with endometriosis. Please cite two recent papers which studied this particular problem (PMID: 35819491 and PMID: 34718292). 

Response 1: We added the influence of endometriosis on impaired life quality of patients in introduction section (page 1 line 35 ~ 39). Two recent papers were also cited.

Point 2: Methodology and Results are clearly presented although I believe that #ENZIAN classification will be of more interest due to its comprehensive nature. This is why I suggest that it is important to include this classification in Discussion section. Please compare its advantages and limitations over ASRM which is used in your study.

Response 2: We compared the advantages and limitations of ENZIAN classification over ASRM system in discussion (page 11 line 394 ~ 397).

Point 3: Please clarify strengths and limitations of your study. 

Response 3: We clarified strengths and limitations of our study in page11 line391 ~ 401.

Point 4: Moderate English changes are required (grammar/spell check).

Response 4: We have checked the grammar and spell in our manuscript.

Round 2

Reviewer 3 Report

This article is now suitable for publication in Diagnostics.